

# Seasonal pattern of influenza and the association with meteorological factors based on wavelet analysis in Jinan City, Eastern China, 2013–2016

Wei Su[1],*, Ti Liu[2],*, Xingyi Geng[3] and Guoliang Yang[3]

[1] School of Management Science and Engineering, Shandong University of Finance and Economics, Jinan, Shandong Province, China
[2] Shandong Center for Disease Control and Prevention, Shandong Provincial Key Laboratory of Infectious Disease Control and Prevention, Shandong University Institution for Prevention Medicine, Jinan, Shandong Province, China
[3] Jinan Center for Disease Control and Prevention, Jinan, Shandong Province, China
* These authors contributed equally to this work.

## ABSTRACT

**Background:** Influenza is a disease under surveillance worldwide with different seasonal patterns in temperate and tropical regions. Previous studies have conducted modeling of influenza seasonality using climate variables. This study aimed to identify potential meteorological factors that are associated with influenza seasonality in Jinan, China.

**Methods:** Data from three influenza sentinel hospitals and respective climate factors (average temperature, relatively humidity (RH), absolute humidity (AH), sunshine duration, accumulated rainfall and speed of wind), from 2013 to 2016, were collected. Statistical and wavelet analyses were used to explore the epidemiological characteristics of influenza virus and its potential association with climate factors.

**Results:** The dynamic of influenza was characterized by annual cycle, with remarkable winter epidemic peaks from December to February. Spearman's correlation and wavelet coherence analysis illuminated that temperature, AH and atmospheric pressure were main influencing factors. Multiple wavelet coherence analysis showed that temperature and atmospheric pressure might be the main influencing factors of influenza virus A(H3N2) and influenza virus B, whereas temperature and AH might best shape the seasonality of influenza virus A(H1N1) pdm09. During the epidemic season, the prevalence of influenza virus lagged behind the change of temperature by 1–8 weeks and atmospheric pressure by 0.5–3 weeks for different influenza viruses.

**Conclusion:** Climate factors were significantly associated with influenza seasonality in Jinan during the influenza epidemic season and the optional time for influenza vaccination is before November. These finding should be considered in influenza planning of control and prevention.

Corresponding author
Wei Su, suwei@sdufe.edu.cn

## INTRODUCTION

Influenza remains a global public health concern, causing 3–5 million severe illnesses and 291–646 thousand deaths (*Iuliano et al., 2018*). Influenza epidemics show distinct seasonality pattern, particularly in winter in temperate areas and a more diverse behavior in tropical and subtropical areas, where influenza displays semi-annual or annual epidemic cycles and year-round activity (*Tamerius et al., 2013*). Moreover, the durations and transmission of influenza outbreaks vary among periods (*Onozuka & Hagihara, 2015*). *Azziz Baumgartner et al. (2012)* showed that seven of 47 (15%) temperate countries presented two or three influenza epidemic period every year and year-round activity in 3 of 43 (7%) temperate countries. Thus, accurately documenting the dynamics of influenza epidemics and understanding its epidemic patterns are of great importance not only to scientific interest, but also public health (*Thai et al., 2015*).

The seasonality of influenza epidemics is associated with many factors, including virus mutation, people susceptibility, climate and environmental changes. Among all of the factors, climate plays the most important role in influenza seasonality (*Lofgren et al., 2007*; *Lowen & Steel, 2014*; *Tamerius et al., 2011*). Experimental studies in guinea pigs showed that the influenza is stable at low relative humidity (RH) and relatively unstable at intermediate RH, with absolute humidity (AH) impacting the influenza virus viability and transmission. *Yang, Elankumaran & Marr (2012)* proposed that the effect of RH on influenza stability may be due to changes in salt concentration within droplets. Influenza virus is also more stable at low temperatures, which may be due to an increased virus half-life at low temperatures, decreased activities of proteases and cooler epithelial surface (*Lowen & Steel, 2014*). Furthermore, temperature and RH influence the innate defense of host nasal epithelia and the production of infectious bio-aerosols, thereby shaping influenza seasonality. Some studies have hypothesized that influenza virus stability increases with reduced sun activity (*Hope-Simpson, 1981*).

To evaluate whether climate factors shape the seasonality of influenza, epidemiological studies comparing influenza seasonality to metrological conditions have been performed, showing that lower temperature and AH increase influenza virus survival and transmission in temperate regions (*Shaman & Kohn, 2009*; *Shaman et al., 2010*; *Shaman, Goldstein & Lipsitch, 2011*). *Tamerius et al. (2013)* and *Bloom-Feshbach et al. (2013)* showed that the onset of influenza epidemics is associated with the cold-dry and humid-rainy condition in temperate and tropical areas, respectively. After analyzing the relative reports of 11 locations in different climate regions, *Chong et al. (2019)* found that temperature and AH were the major determinant to influenza A and influenza B and RH was less consistent with influenza B activity, while *Peci et al. (2019)* reported the negative relationship of both AH and temperature with influenza virus A and B and a controversial effect of RH on influenza virus A and B. Although influenza patterns have been well described in developed temperate regions, those involving developing temperate areas are limited. *Yu et al. (2013)* showed that minimum temperature, the hours of sunshine and maximum rainfall are closely related with influenza seasonality in China, but AH was not taken into consideration and the influenza surveillance data from April to
September in northern China was missing. Several studies teams demonstrated the negative association between influenza virus with temperature and AH in Beijing and Gansu Province, North of China, respectively (*Cao et al., 2010*; *Sun, Sun & Xiao, 2018*; *Yang et al., 2019*). However, studies on the relationship between different influenza virus types or subtypes and climate factors are limited. Therefore, understanding the influence of climate factors is vitally important to learn the influenza seasonality pattern and consequently implement prevention and control strategies.

In this study, we analyzed the seasonality of influenza and explored various climate factors, including AH, RH, temperature, atmosphere, the hours of sunshine, wind speed and accumulated rainfall as potential drivers to the seasonality of laboratory-confirmed influenza cases in Jinan City, the capital of Shandong Province, eastern of China, from 2013 to 2016. Jinan is a temperate city with seven million people and three sentinel hospitals for the surveillance of influenza. This detailed analysis of influenza pattern aim to provide evidence to foster strategies to prevent and control influenza in future years.

## METHODS

### Influenza surveillance data

Epidemiological and virology data of three national influenza surveillance sentinel hospitals was collected from Jinan Center for Disease Control and Prevention for the study period of 2013–2016, including QiLu Children's Hospital of Shandong University, the No. 4 Hospital of Jinan and the No. 6 Hospital of Jinan (The People's Hospital of Zhang-QiuArea, Jinan) located in different three zones and represented about a third of the total population. For each hospital, at least 20 nasopharyngeal swab specimens were collected each week from patients with influenza-like illness (ILI), which was defined as an outpatient of any age with an acute respiratory infection syndrome with fever ≥38 °C and cough or sore throat. All collected samples were tested for influenza virus by virus isolation in Madin–Darby canine kidney cells. The types (influenza A and B), subtypes (A(H3N2) and A(H1N1)pdm09), and lineages of influenza B (Yamagata and Victoria) were identified by haemagglutination assay and haemagglutination inhibition (HI) test according to National Influenza Surveillance Protocol published by Chinese Health commission of the People's Republic of China. In this study, we used the weekly positive rate of different influenza virus types or subtypes as influenza proxy (Positive rate = Number of positive specimens/Total number of specimens tested per week ×100%) to describe the seasonality of influenza according to *Tamerius et al. (2011)* and *Liu et al. (2017)*.

### Climate data

Weekly meteorological data from 2013 to 2016 from Jinan were obtained from Meteorological Science Data Sharing Service System, including weekly average values of temperature (°C), atmospheric pressure (hPa), RH (%), wind speed (m/s), accumulated rainfall (mm), and sunshine duration (h). The weekly average AH (g/m$^3$) was calculated based on the data of average temperature and RH using the formula described by *Xiao et al. (2012)*.

## Statistical methods

A descriptive analysis was used to reveal the characteristics of influenza epidemics and climate factors. To characterize the epidemics of influenza, continuous wavelet transform (CWT) was applied to uncover the time frequencies of influenza virus *Cazelles et al. (2007)*. This technique has been widely used to explore the temporal and spatial variations of different infectious diseases, including influenza (*Xiao et al., 2012*; *Liu et al., 2017*). Wavelet transform can divide the time series into a family of wavelets using dilated and translated functions called "mother wavelets". In this study, the Morlet wavelet was selected as the mother, which can be well localized in scales and in high-frequency resolution and is regarded as an efficient means of detecting and analyzing the time series (*Ng & Chan, 2012*). Wavelet analysis is represented by a colored contour. The 5% significant level against red noise is shown as a thick black curve. The cone of influence, which indicates the region affected by edge effects, is shown with a lighter shade. The color code for power ranges from pink (low power) to white (high power).

To explore the potential relationship between influenza and climate variables, Spearman's correlation was conducted to determine the correlation between different climate factors and influenza virus types or subtypes.

Wavelet transform coherence (WTC) was used to investigate the possible inter-connectedness between influenza and climate variables according to the method described by *Grinsted, Moore & Jevrejeva (2004)*. The WTC shows the significant coherence against red noise in time-frequency space, which can depict the significant covariance at specific periods (frequencies) and phase shift between two time-series. The phase difference between the two series is indicated by arrows. If the arrow points to the right (left), the series are positively (negatively) correlated, that is, they are in the in-phase or the anti-phase, respectively and if the arrow points straight down (up), the first series leads the other by 90° (vice versa). For the detail information, please refer to the method described by *Grinsted, Moore & Jevrejeva (2004)*.

Finally, to exploit the possible relationship between influenza and climate factors, multiple wavelet coherence (MWC) was applied. MWC is a technique that is similar to multiple correlations and helps to seek the resulting wavelet coherence of two independent variables on a dependent one at a given time and frequencies (*Grinsted, Moore & Jevrejeva, 2004*; *Ng & Chan, 2012*). The continuous wavelet analysis, wavelet transform coherence and MWC were performed based on a library of MATLAB functions provided by *Grinsted, Moore & Jevrejeva (2004)* (http://noc.ac.uk/business/marine-data-products/cross-wavelet-wavelet-coherence-toolbox-matlab).

# RESULTS

## Descriptive statistics

### Seasonality and periodicity of influenza virus in Jinan

A total of 9,170 ILI samples were collected and 914 laboratory-confirmed cases (9.94%) were identified by virus isolation from 2013 to 2016, including 370 influenza virus A(H1N1)pdm09, 299 influenza virus A(H3N2) and 245 influenza virus B (219B/Yamagata

**Table 1 Description of weekly climatic factors in Jinan, 2013–2016.**

| Variables | Minimum | Maximum | Mean | Standard deviation |
|---|---|---|---|---|
| Atmospheric pressure (hPa) | 979.4 | 1,013.4 | 996.5 | 8.5 |
| Mean temperature (°C) | −6.4 | 30.6 | 15.3 | 9.9 |
| Relative humidity (%) | 23.3 | 91.7 | 56.4 | 14.6 |
| Absolute humidity (g/m$^3$) | 1.5 | 17.9 | 7.5 | 4.5 |
| Wind speed (m/s) | 1.0 | 4.5 | 2.5 | 0.6 |
| Sunshine duration (h) | 5.3 | 77.9 | 37.5 | 17.1 |
| Rainfall (mm) | 0 | 102.4 | 5.1 | 13.7 |
| Influenza A(H1N1)pdm09 virus (number) | 0 | 37 | 1.8 | 5.5 |
| Influenza A(H3N2) virus (number) | 0 | 24 | 1.4 | 4.0 |
| Influenza virus B (number) | 0 | 28 | 1.2 | 3.3 |

lineage and 26 B/Victoria lineage) (Table 1). A(H1N1)pdm09 and A(H3N2) were detected from 2013 to 2016, accounting for 73.19% of the total positive influenza virus cases. Figure 1A shows the seasonal distribution of different influenza types, subtypes and lineages during the periods, which clearly indicated that influenza had seasonality, with one epidemic peak during the winter season. Figure 1B revealed the wavelet power spectrum of the influenza virus. A high-power spectrum indicated dominant frequency- and time-specific periodicity. Although annual influenza epidemic cycles were identified in the 32–64 week band (2013–2016) with high power, influenza virus also showed a distinct epidemic peak with high power in the 8–14 and 0–4 week bands in 2014–2015 and 2014–2016 ($p < 0.05$), respectively.

Figure 2 demonstrates the seasonality of influenza virus A and B in Jinan. Similar to the total influenza activity, A(H3N2 ), A(H1N1)pdm09 and influenza virus B showed an annual cycle with a peak in winter (Figs. 2A, 2C and 2E). Based on CWT, A(H3N2) exhibit a statistically significant periodicity during 2014–2015 with higher power in the 0–4 week band (Fig. 2B) and A(H1N1)pdm09 had two statistically significant regions in 2015–2016 with higher power in the 0–4 and 8–16 week bands (Fig. 2D). Influenza virus B exhibited a statistically significant periodicity with three regions of high powers in the 0–4 week band during 2014–2016 and one high power in the 6–24 week band in 2014–2015 (Fig. 2F).

### Seasonality and periodicity of climate factor in Jinan
Table 1 showed the average values of the weekly climate factors: atmospheric pressure, temperature and RH, AH, wind speed, sunshine duration and accumulated rainfall during the study period. Figure 3 demonstrates the weekly variation in climate factors with clear seasonal and periodic characteristics. Atmospheric pressure and influenza positive rate showed similar patterns, but average temperature and AH exhibited different variations.

### Spearman's correlation
Table 2 shows the relative Spearman's correlation of different influenza viruses and climate factors. Spearman's correlation analysis shows that weekly average atmospheric pressure is positively correlated with influenza. In contrast, weekly average values of

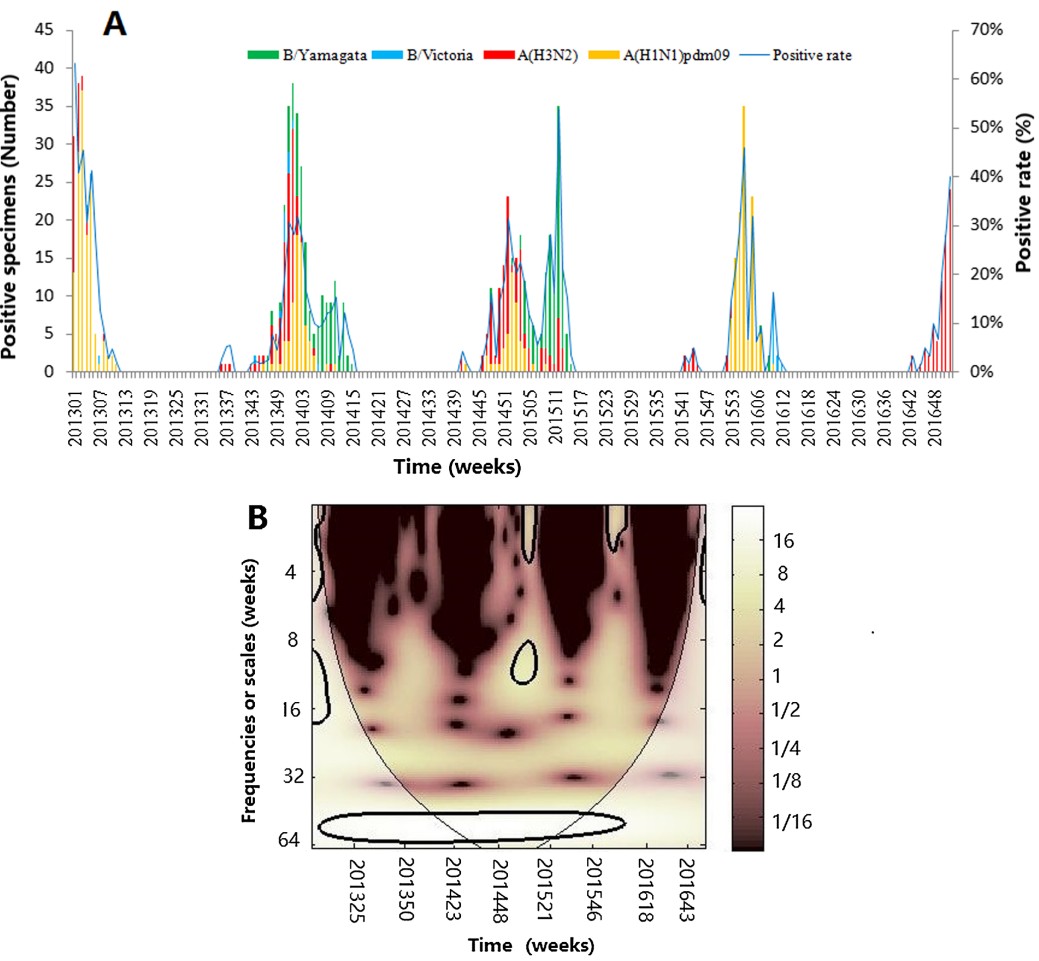

**Figure 1 The composite influenza virus activity in Jinan, Eastern China, 2013–2016.** (A) The weekly positive rates of all influenza viruses combined and week distribution of influenza virus A (A(H3N2) and A(H1N1)pdm09) and influenza virus B (B/Yamagata and B/Victoria) (B) wavelet power spectrum of the positive rate of all influenza viruses combined. The 5% significant level against red noise is shown as a thick black curve. The cone of influence, which indicates the region affected by edge effects, is shown with a lighter shade. The color code for power ranges from pink (low power) to white (high power). X-axis denotes the time and Y-axis represents frequency periods.

temperature, AH, RH and accumulated rainfall are negatively associated with influenza. The Spearman's correlation between climate factors and three influenza virus types or subtypes demonstrated medium association for atmospheric pressure, temperature and AH and weak relationship for RH and accumulated rainfall. Influenza has no correlation with the weekly average wind speed and the weekly average sunshine duration ($p > 0.05$), except for A(H3N2) with the weekly average wind speed. Based on these observations, atmospheric pressure, temperature, AH, RH, accumulated rainfall and wind speed (only for A(H3N2)) were selected for further analysis.

### Wavelet transform coherence

The results of WTC of influenza and different climate factors are shown in Figs. 4–6. WTC provided information on whether two time series are linearly correlated or co-moved at a

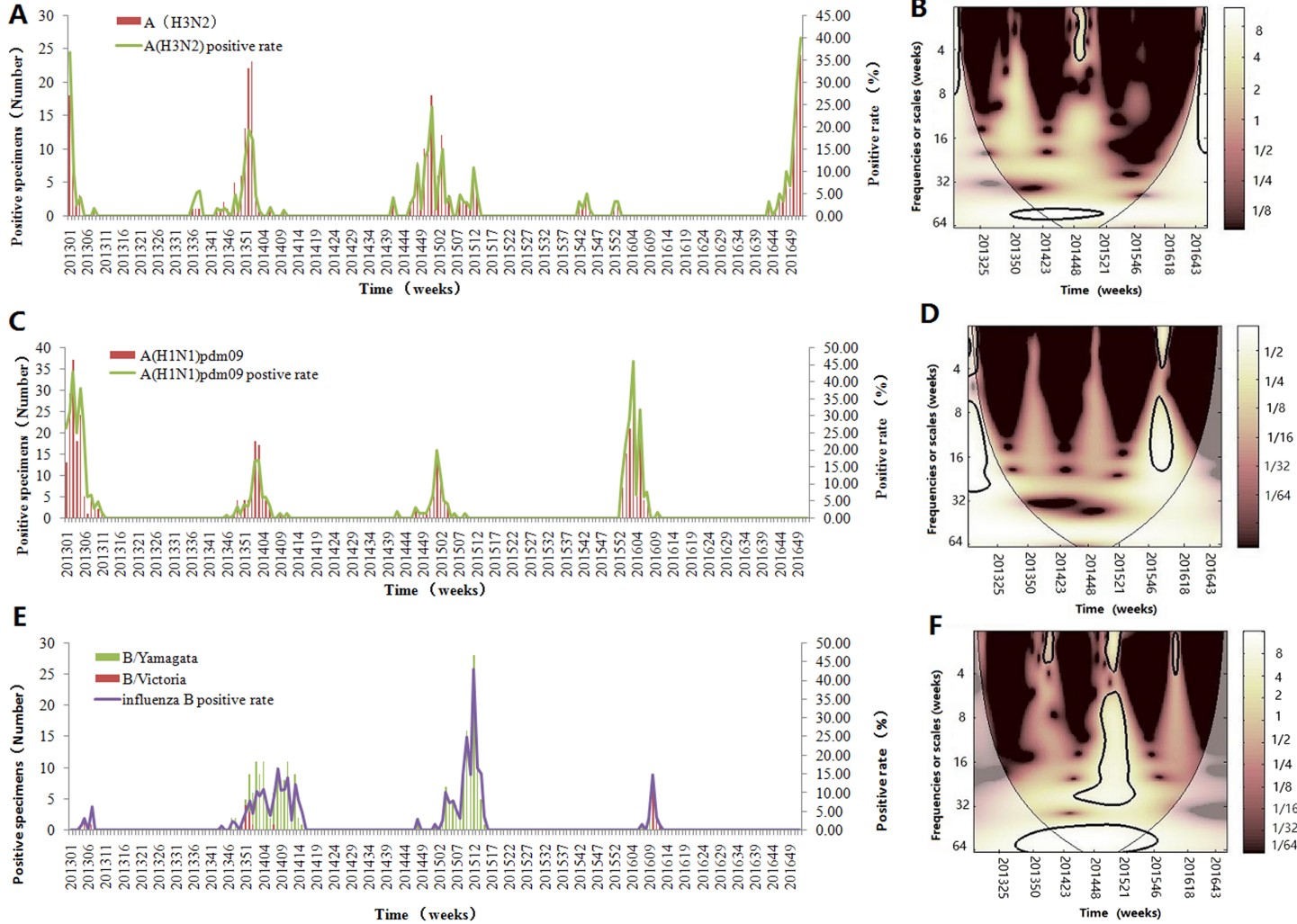

**Figure 2 Seasonal distribution and wavelet power spectrum of influenza types or subtypes.** (A) The weekly positive rate and week distribution of influenza virus A(H3N2). (B) Wavelet power spectrum of the positive rate of influenza virus A(H3N2). (C) The weekly positive rate and week distribution of influenza virus A(H1N1)pdm09. (D) Wavelet power spectrum of the positive rate of influenza virus A(H1N1)pdm09. (E) The weekly positive rate and week distribution of influenza virus B. (F) Wavelet power spectrum of the positive rate of influenza virus B. The 5% significant level against red noise is shown as a thick black curve. The cone of influence, which indicates the region affected by edge effects, is shown with a lighter shade. The color code for power ranges from pink (low power) to white (high power). X-axis denotes the time and y-axis represents frequency periods.

particular time and frequency, in which the whiter the color, the higher the correlation. Statistically significant relationships are highlighted by a thick black curve around the significant regions. The cone of influence is shown with a lighter shade black line. The color code for power ranges from pink (low power) to white (high power). The orientation of the arrows represents the lag or lead relationship. The down arrows show climate factor is leading. The up arrows denote influenza is leading.

For influenza virus A (H3N2), the result of WTC shows that it is statistically significantly associated with some climate factors in some periods (Fig. 4). The atmospheric pressure was significantly correlated at the period of approximately 26 week band (2013–2014) ($p < 0.05$), with arrows pointing to the left, indicating that both

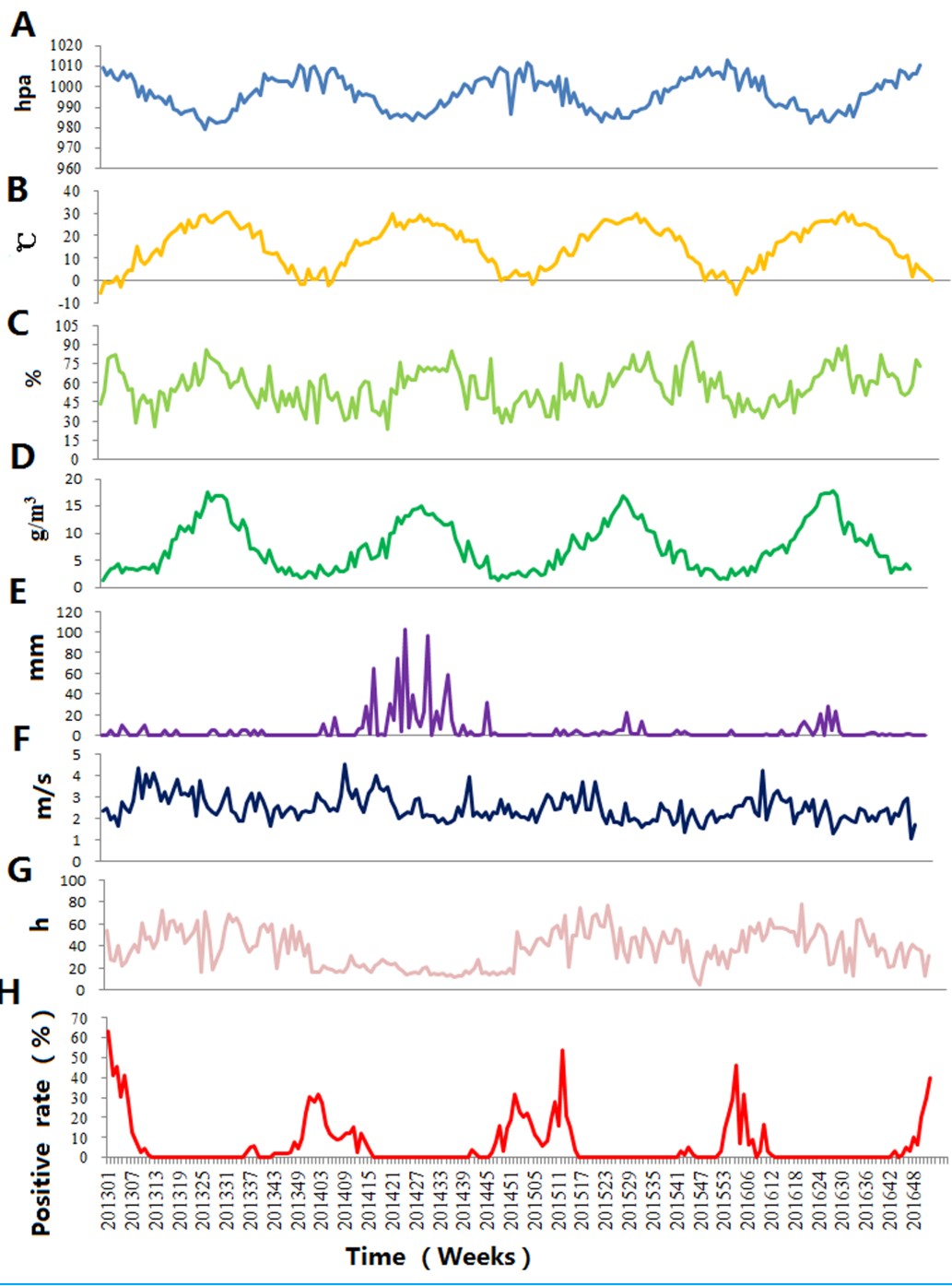

**Figure 3** **The time-series results of the positive rate of influenza virus and climate factors in Jinan, 2013–2016.** (A) Atmospheric pressure, (B) average temperature, (C) relative humidity, (D) absolute humidity, (E) accumulated rainfall, (F) wind speed, (G) sunshine duration and (H) influenza positive rate.                                                                         

are negatively correlated during 2013–2014. For temperature, A(H3N2) was significantly correlated at the period of approximately 26 week band (2013–2014) and the arrows pointed down, indicating that the changes of temperature leads the prevalence of influenza

**Table 2 The correlation between positive rate of different influenza virus types or subtypes and different climatic factors from 2013 to 2016.**

| Variables | Atmospheric pressure | Mean temperature | Relative humidity | Absolute humidity | Wind speed | Sunshine duration | Rainfall |
|---|---|---|---|---|---|---|---|
| Influenza A(H1N1)pdm09 | 0.521**[a] | −0.632** | −0.245** | −0.623** | −0.020 | −0.132 | −0.240** |
| Influenza A(H3N2) virus | 0.539** | −0.522** | −0.139** | −0.487** | −0.164* | −0.122 | −0.285** |
| Influenza virus B | 0.377** | −0.450** | −0.292** | −0.494** | 0.097 | −0.063 | −0.277** |

Notes:
[a] The strength of the association is interpreted as weak when coefficients are in the range of 0.1–0.3, medium when the coefficients are in the range of 0.3–0.7 and strong when the coefficients are in the range of 0.7–1.0, according to the report of Chong et al. (2019).
* $p < 0.05$.
** $p < 0.01$.

by 8 weeks. For RH, we also observed one high-power spectrum in the 8–12 week band (2014–2015) with anti-phase, indicating that both are negatively correlated during the period. At the same time, in the 32–64 week timescale, we found a strong, but not statistically significant at the 5% level, correlation during the study period between A(H3N2) and different climate factors, showing that atmospheric pressure is positively correlated and the other climate factors except for wind speed were negatively correlated with A(H3N2) on the annual cycle during the periods.

For influenza virus A(H1N1)pdm09, we also observed a statistically significant association with atmospheric pressure, temperature, RH and AH on the short or medium scales (Fig. 5). For atmospheric pressure, A(H1N1)pdm09 was significantly negatively correlated at periods of approximately 26 week band (2014) and 1–12 week band (2014–2015). The power was also observed in the 3–6 week band in 2013–2014 with arrows pointing down, which illuminated that the prevalence of influenza virus lags by approximately 0.5–1 week. In terms of temperature, in the 26 week band (2013–2014), A(H1N1)pdm09 was negatively correlated. For the periods of 6–10 week band in 2015–2016, we observed a negative relationship, in which the change of temperature leads the prevalence of influenza virus by approximately 2–3 weeks. But for the bands of the 6–8 (2013–2014) and 6–12 (2014–2015) week, positive correlations were observed. At the same time, a significant period of 4–6 week band (2015–2016) of AH were also observed, showing that the change of AH leads the prevalence of influenza virus by 2–3 weeks during the periods. For RH, we only observed a statistically significant region in the 8–10 week band (2014–2015), with influenza virus leading. Similar to A (H3N2), on the 1 year scale, atmospheric pressure was positive and the other climate factors except for rainfall were negative with A(H1N1)pdm09 during the periods.

For influenza virus B, we observed statistically significant correlation with atmospheric pressure in the approximately 26 week band (2014–2015) and 6–10 week band (2015–2016) (Fig. 6A), with the change of atmospheric pressure leading the prevalence of influenza by 3 weeks and 1–1.5 weeks, respectively. For temperature, influenza virus B was significantly correlated at periods of approximately 6–12 week band (2013–2014), showing that temperature leads influenza by 1–2 weeks. Similarly, atmospheric pressure was positively correlated and the other climate factors were negatively correlated with influenza virus B on the one-year scale (Fig. 6).

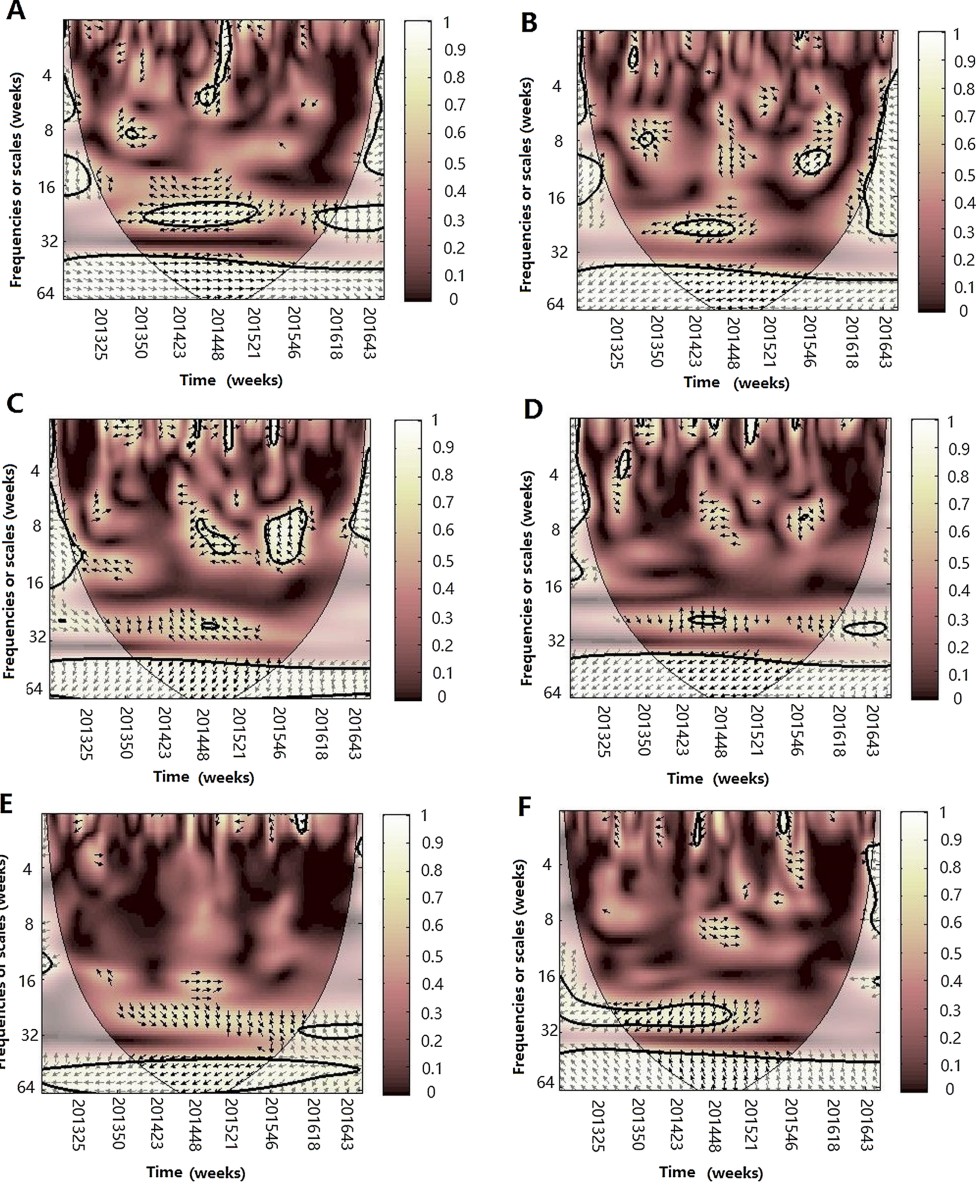

**Figure 4 The wavelet transform coherence of influenza virus A(H3N2) and different climate parameters.** Wavelet transform coherence is represented by a colored contour: the whiter the color is, the higher the local correlation in the time-frequency periods space (with time on the *x*-axis and frequencies on the *y*-axis). The matching of colors and correlation levels is represented by the scale on the right hand side of the upper graph. The 5% significant level against red noise is shown as a thick black curve. The cone of influence, which indicates the region affected by edge effects, is shown with a lighter shade black line. The color code for power ranges from pink (low power) to white (high power). The phase difference between the two series is indicated by arrows. Arrows pointing to the right mean that the variables are in phase. Arrows pointing to the left mean that the variables are out of phase. The down arrows show that climate factor is leading. The up arrows mean that influenza virus is leading. In phase indicate that variables will be having cyclical effect on each other and out of phase or anti-phase shows that variable will be having ant-cyclical effect on each other. (A) Influenza virus A(H3N2) and atmospheric pressure, (B) influenza virus A(H3N2) and average temperature, (C) influenza virus A(H3N2) and relatively humidity, (D) influenza virus A(H3N2) and absolute humidity, (E) influenza virus A(H3N2) and rainfall and (F) influenza virus A(H3N2) and wind speed.

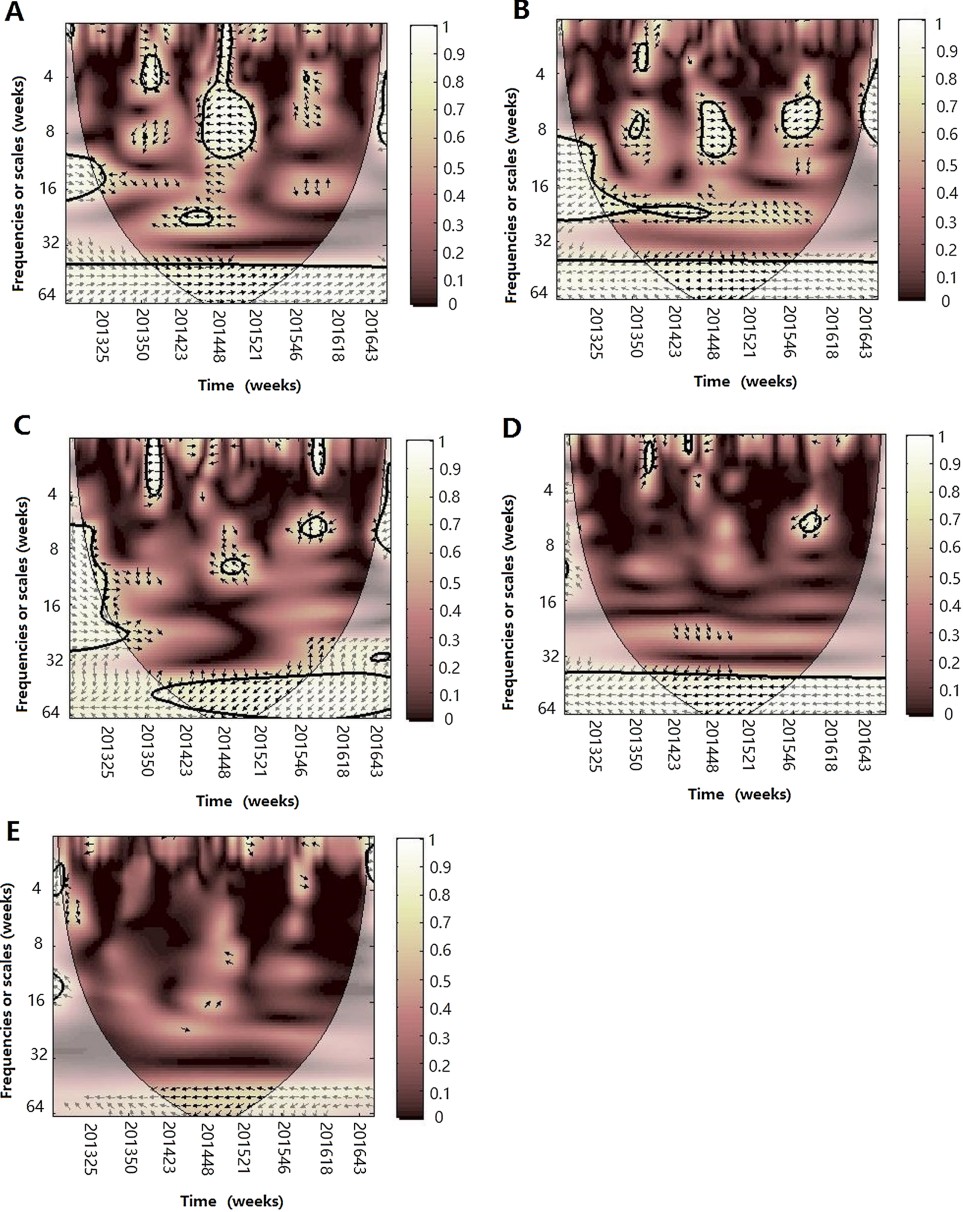

**Figure 5 The wavelet transform coherence of influenza virus A(H1N1)pdm09 and different climate parameters.** Wavelet transform coherence is represented by a colored contour: the whiter the color is, the higher the local correlation in the time-frequency periods space (with time on the *x*-axis and frequencies on the *y*-axis). The matching of colors and correlation levels is represented by the scale on the right hand side of the upper graph. The 5% significant level against red noise is shown as a thick black curve. The cone of influence, which indicates the region affected by edge effects, is shown with a lighter shade black line. The color code for power ranges from pink (low power) to white (high power). The phase difference between the two series is indicated by arrows. Arrows pointing to the right mean that the variables are in phase. Arrows pointing to the left mean that the variables are out of phase. The down arrows show that climate factor is leading. The up arrows mean that influenza virus is leading. In phase indicate that variables will be having cyclical effect on each other and out of phase or anti-phase shows that variable will be having ant-cyclical effect on each other. (A) Influenza virus A(H1N1)pdm09 and atmospheric pressure, (B) influenza virus A(H1N1)pdm09 and average temperature, (C) influenza virus A(H1N1)pdm09 and relatively humidity, (D) influenza virus A(H1N1)pdm09 and absolute humidity and (E) influenza virus A(H1N1)pdm09 and rainfall.

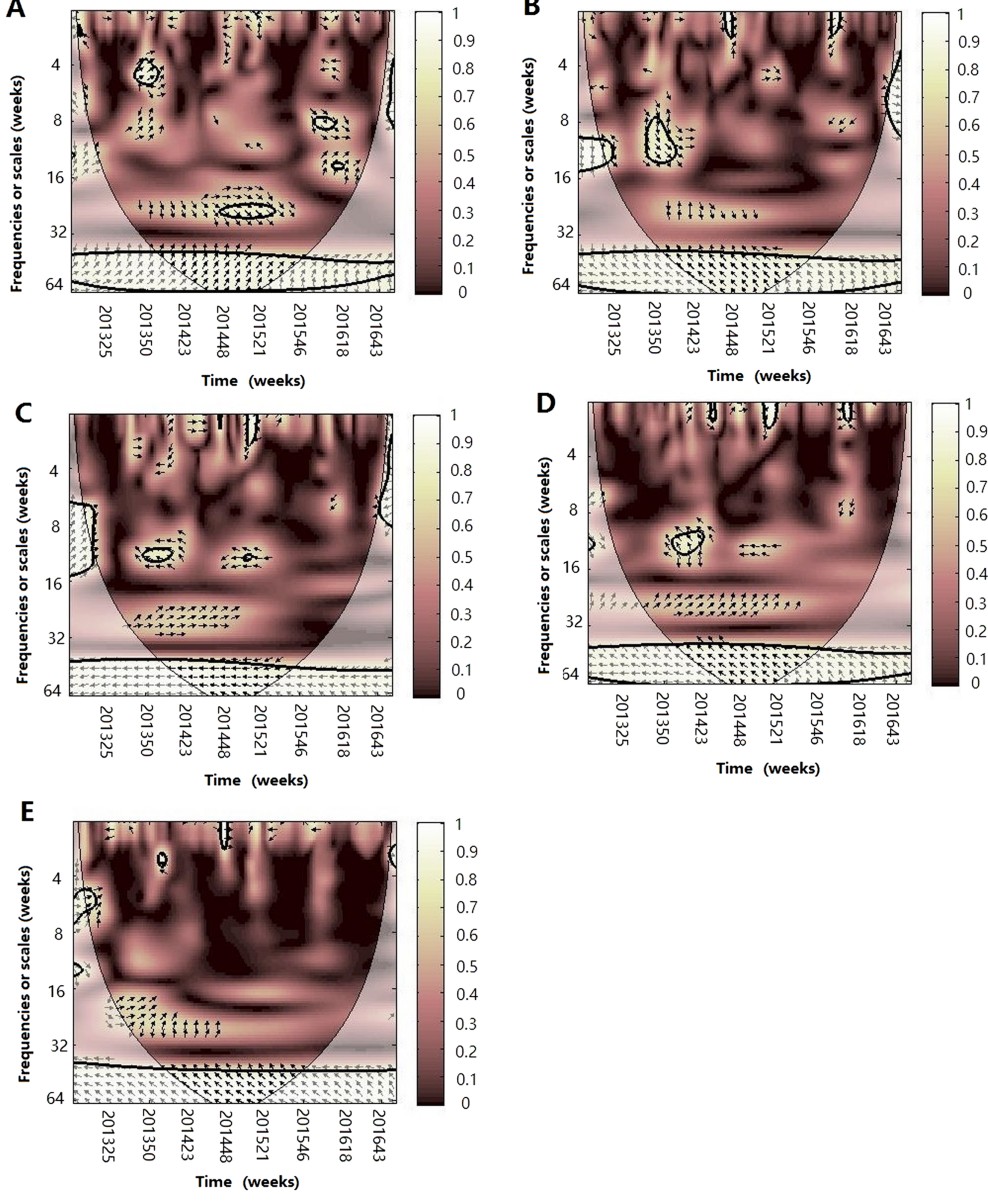

**Figure 6 The wavelet transform coherence of influenza virus B and different climate parameters.**
Wavelet transform coherence is represented by a colored contour: the whiter the color is, the higher
the local correlation in the time-frequency periods space (with time on the *x*-axis and frequencies on the
*y*-axis). The matching of colors and correlation levels is represented by the scale on the right hand side of
the upper graph. The 5% significant level against red noise is shown as a thick black curve. The cone of
influence, which indicates the region affected by edge effects, is shown with a lighter shade black line.
The color code for power ranges from pink (low power) to white (high power). The phase difference
between the two series is indicated by arrows. Arrows pointing to the right mean that the variables are in
phase. Arrows pointing to the left mean that the variables are out of phase. The down arrows show that
climate factor is leading. The up arrows mean that influenza virus is leading. In phase indicate that
variables will be having cyclical effect on each other and out of phase or anti-phase shows that variable
will be having ant-cyclical effect on each other. (A) Influenza virus B and atmospheric pressure,
(B) influenza virus B and average temperature, (C) influenza virus B and relatively humidity, (D) influ-
enza virus B and absolute humidity and (E) influenza virus B and rainfall.

*Multiple wavelet coherence analysis*

After WTC, temperature, humidity (AH and RH) and atmospheric pressure were involved with MWC analysis. The combined impact of any two of four different climate factors and influenza virus can be analyzed using MWC squared. The plots of MWC of three influenza virus types or subtypes are shown in Figs. 7–9. Through MWC by comparing different combinations of the climate factors, the results show that not only the short-term (8–14 week band), but also the medium-term (26 week band) and the long-term trend relationships or co-movement between climate factors and influenza virus types or subtypes are correlated. For example, the MWC (influenza A(H3N2)-temperature-RH) plot in Fig. 7C shows that three time series are correlated significantly and share the same coherent areas in the 26 week band (2013–2016) and in the 6–12 week period (2013–2016). Thus, based on Figs. 7–9, we concluded that the combination of temperature and atmospheric pressure, temperature and AH and temperature and atmospheric pressure were the main potential influencing factors of influenza virus B, A(H1N1)pdm09 and A(H3N2), respectively.

## DISCUSSION

In this study, we exploited influenza seasonality and the potential relationship between influenza seasonality and different climate factors. Through descriptive analysis and wavelet analysis, we found that influenza virus exhibited an annual epidemic cycle with remarkable winter seasonality in Jinan and concluded that the potential relationships or co-movement of influenza virus types or subtypes and different climate factors were quite correlated in Jinan from 2013 to 2016.

The laboratory-confirmed data is thought to be a better indicator to depict the influenza seasonality (*Tamerius et al., 2011*; *Liu et al., 2017*), compared with influenza-like illness which was used to represent the level of influenza activity in some studies (*Azziz Baumgartner et al., 2012*; *Thai et al., 2015*; *Tamerius et al., 2011*). Thus, in our study, the weekly positive rate of influenza virus was used to precisely illustrate the epidemic and seasonality of influenza. The description statistics and CWT analysis indicated that influenza had an annual cycle with regular peak seasonality in winter season and was scarce in the summer and autumn seasons in Jinan City, which was in accordance with the findings of previous reports in temperate regions (*Wu et al., 2016*; *Caini et al., 2016*, *2017*). The CWT analysis further illuminated that the onset of epidemic of influenza always began in December and the epidemic peak time could last for 8–14 weeks in every surveillance year (Fig. 1B). Unlike other studies in which influenza had two or three peaks in a temperate zone (*Azziz Baumgartner et al., 2012*), we only found an obvious epidemic peak in Jinan. By analyzing the time series of each influenza virus type or subtype, the descriptive analysis and CWT analysis also showed that three influenza viruses were detected every year with one or two subtypes predominated in winter and the predominant subtype changed with the years. In the early stage of 2013, A(H1N1)pdm09 was the main epidemic virus. In 2013–2015 surveillance years, influenza virus B and A(H3N2) were the main epidemic strains with few A(H1N1)pdm09 detected. In 2015–2016 surveillance year, the predominant virus was changed to A(H1N1)pdm09

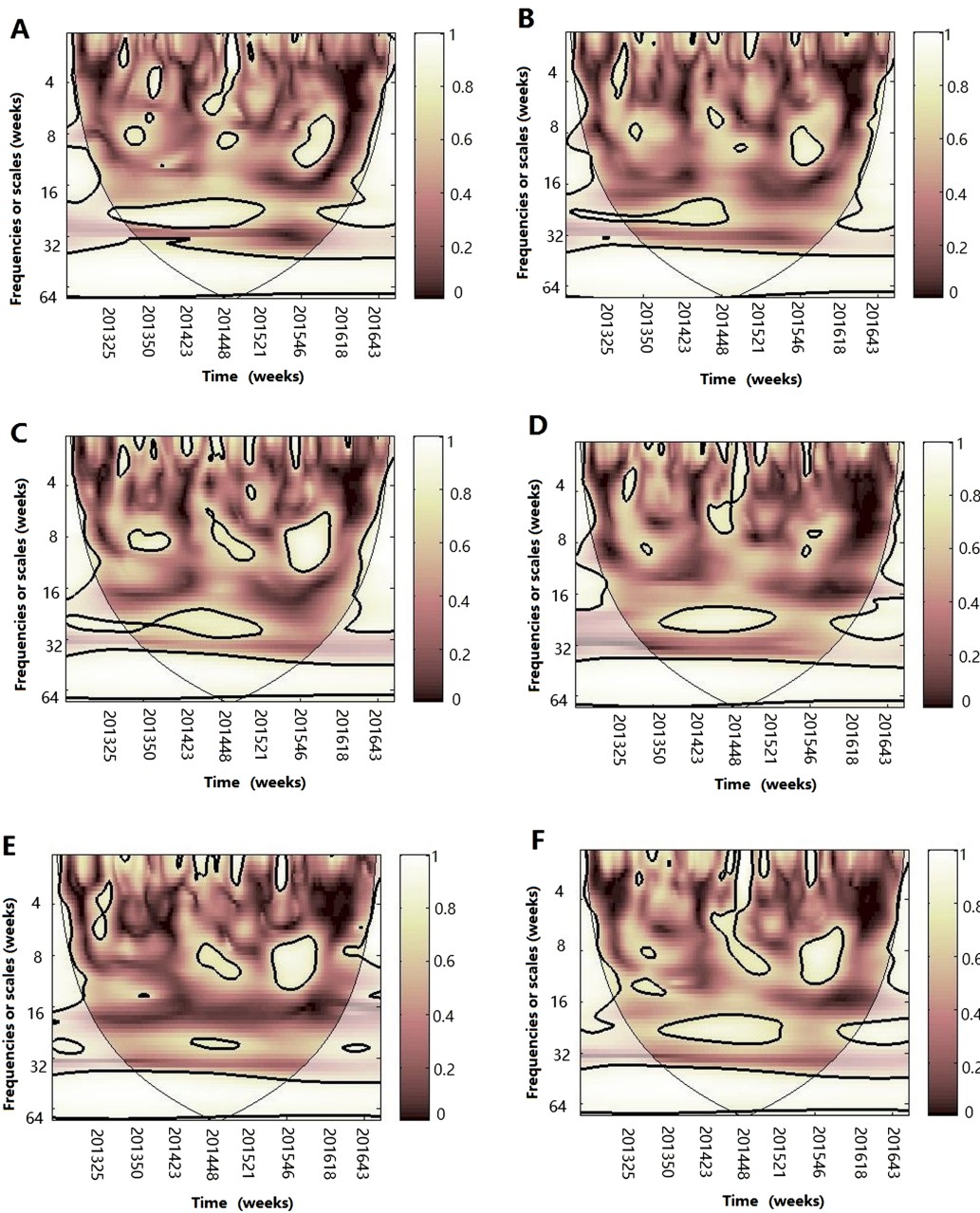

**Figure 7 Multiple wavelet coherence of influenza A(H3N2) virus and climate factors.** The 5% significant level against red noise is shown as a thick black curve. The cone of influence, which indicates the region affected by edge effects, is shown with a lighter shade. The color code for power ranges from pink (low power) to white (high power). (A) Influenza virus A(H3N2), temperature and atmospheric pressure, (B) influenza virus A(H3N2), temperature and AH, (C) influenza virus A(H3N2), temperature and RH, (D) influenza virus A(H3N2), AH and atmospheric pressure, (E) influenza virus A(H3N2), AH and RH and (F) influenza virus A(H3N2), RH and atmospheric pressure.

and A(H3N2) become the predominant virus in the last month of 2016. Therefore, understanding the influenza seasonality suggested that the optional time of influenza vaccination programs was from September to November in Jinan, which provided immune

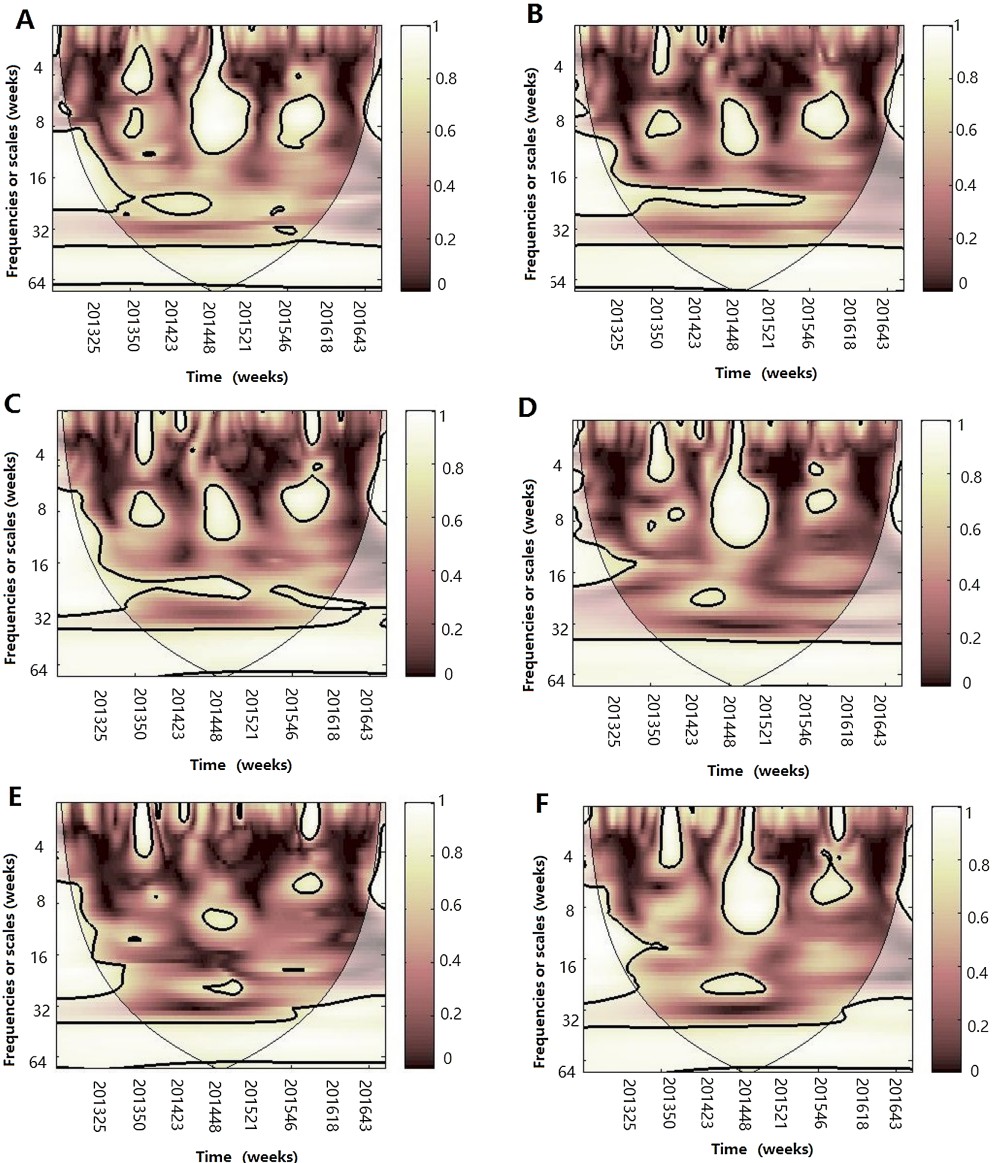

**Figure 8 Multiple wavelet coherence of influenza A(H1N1)pdm09 virus and climate factors.** The 5% significant level against red noise is shown as a thick black curve. The cone of influence, which indicates the region affected by edge effects, is shown with a lighter shade. The color code for power ranges from pink (low power) to white (high power). (A) Influenza virus A(H1N1)pdm09, temperature and atmospheric pressure, (B) influenza virus A(H1N1)pdm09, temperature and AH, (C) influenza virus A(H1N1)pdm09, temperature and RH, (D) influenza virus A(H1N1)pdm09, AH and atmospheric pressure, (E) influenza virus A(H1N1)pdm09, AH and RH and (F) influenza virus A(H1N1)pdm09, RH and atmospheric pressure.

protection in the epidemic peak time. Otherwise, the vaccine effectiveness wanes over time (*Kissling et al., 2013*; *Pebody et al., 2013*).

The seasonality of influenza indicated a peak in winter and the predominant influenza virus changed with the years. As for the mechanism of the seasonality and alternation of influenza, host susceptibility, virus mutation and climate factors may have driven influenza virus circulation activity. Therefore, we exploited the potential association

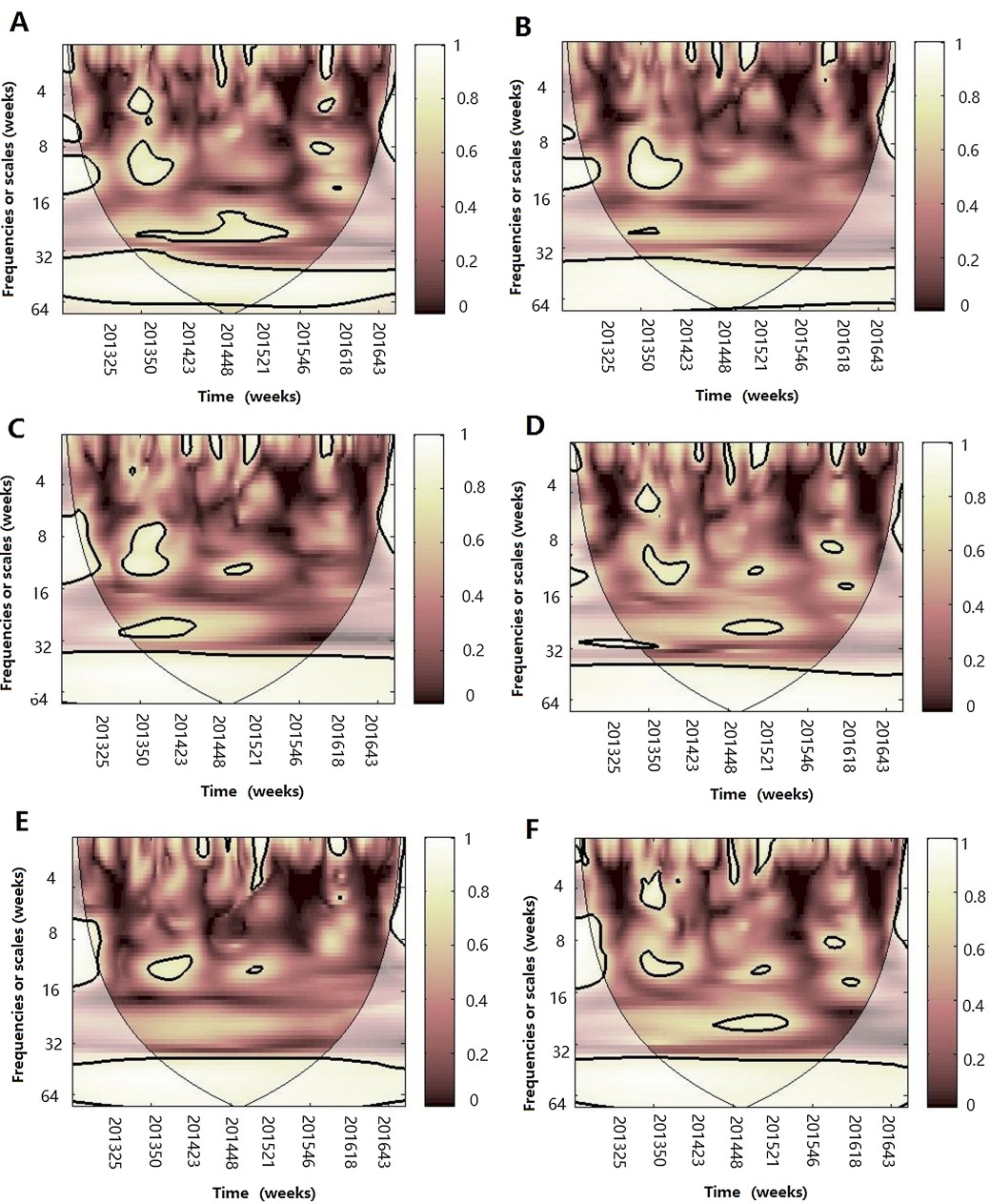

**Figure 9 Multiple wavelet coherence of influenza virus B and climate factors.** The 5% significant level against red noise is shown as a thick black curve. The cone of influence, which indicates the region affected by edge effects, is shown with a lighter shade. The color code for power ranges from pink (low power) to white (high power). (A) Influenza virus B, temperature and atmospheric pressure, (B) influenza virus B, temperature and AH, (C) influenza virus B, temperature and RH, (D) influenza virus B, AH and atmospheric pressure, (E) influenza virus B, AH and RH and (F) influenza virus B, RH and atmospheric pressure.

between influenza virus subtypes and different climate variables through the WTC analysis and MWC analysis. We speculated weekly change in climate variables significantly contributed to dynamic variation of influenza, which was in agreement with the findings of previous studies (*Tamerius et al., 2013*; *Onozuka & Hagihara, 2015*;

*Azziz Baumgartner et al., 2012*; *Chong et al., 2019*). Through WTC analysis and MWC, we concluded that there were statistically significant associations between atmospheric pressure, temperature and humidity (RH and AH) and influenza seasonality.

For weekly atmospheric pressure, we observed that there was positive association with three different influenza virus subtypes in the annual cycle, but this relationship varied with week time scales during 2013–2016, which was consistent with the findings of previous studies in avian influenza viruses (*Li et al., 2015*; *Soebiyanto, Adimi & Kiang, 2010*). On the short- and the medium-term, atmospheric pressure may influence the variations in different influenza virus types, which concluded that atmospheric pressure may be a driver of influenza. The results were verified by MWC analysis, which was consistent with the results of *Sundell et al. (2016)*. Furthermore, during some time periods, the prevalence of influenza virus lags the changes in atmospheric pressure, which may speculate the occurrence of A(H3N2) epidemics. The reason could be that the mucous membrane of nasal cavity was easily broken in dry conditions and high atmospheric pressure in the winter season and virus could easily infect it. However, the detailed mechanism requires further investigation.

Temperature was often reported to be associated with influenza seasonality, especially in warm regions with influenza epidemic peaking in the winter (*Yu et al., 2013*; *Soebiyanto, Adimi & Kiang, 2010*; *Tamerius et al., 2019*). In Jinan, we also observed a negative association between average temperature and three influenza virus subtypes through Spearman's correlation analysis and WTC. Except for the negative relationships, we also observed that during some time periods, changes in temperature lead the prevalence of A(H1N1)pdm09 and influenza virus B by 2–3 weeks (2015–2016) and 1–2 weeks (2013–2014), respectively. Therefore, temperature may be a good predictor of influenza virus epidemics, not only influenza virus A, but also influenza virus B, which was consistent with the results of *Chong et al. (2019)* and *Peci et al. (2019)*. Indeed, most of the studies demonstrated that temperature was a dominant determinant of influenza virus A and B in temperate regions, which was observed in a guinea pig model (*Lowen & Steel, 2014*). Several studies showed that influenza is more stable at low temperature which decreases activities of proteases, reduces the mucus and ciliary movement and inhibits defense and immunity toward infection, thus affects the host susceptibility and increases the transmission of influenza virus in winter (*Lowen & Steel, 2014*; *Peci et al., 2019*). *Lowen & Steel (2014)* further verified that the influenza transmission was most efficient at low temperature (5 °C), the transmission and survival capacity declined as temperature increased from 5 °C to 20 °C and was completely blocked at 30 °C based on the study conducted in guinea pigs. In Jinan, the influenza peak always occurred in December–February with mean temperature at −5 °C to 5 °C, which increased the influenza transmission.

For RH, we observed the weakly negative association with three influenza virus subtypes with Spearman's correlation analysis and WTC in different week bands during various periods and we did not observe any lead correlations with influenza. Similarly, *Noti et al. (2013)* found that influenza virus remains five times more infectious at low RH (7–23%) than at an RH of 43% and above. The mechanism would be explained that the virus aerosol
remains a longer time at low RH and increased the opportunity to infect new hosts (*Sundell et al., 2016*).

In the study, we also observed a negative association between AH and influenza virus. For A(H1N1)pdm09, change in AH leads the prevalence by 2–3 weeks during 2015–2016, which revealed that AH may be a good predictor of A(H1N1)pdm09. For influenza B, we also observed a potential negative correlation during the 8–12 week band during 2013–2014. MWC analysis further demonstrated that AH might influence the seasonality of influenza virus, which coincided with the experimental results in guinea pig and the epidemiology data analysis (*Shaman et al., 2010*; *Shaman, Goldstein & Lipsitch, 2011*; *Lowen & Steel, 2014*). The mechanism may be that the high level of AH deactivate the virus surface because of the denaturing of virus lipoproteins (*Yang, Elankumaran & Marr, 2012*).

Previous studies have reported that the amount of rainfall increases influenza virus activity in tropical and subtropical areas such as Thailand, Guiana in French (*Pica & Bouvier, 2012*; *Mahamat et al., 2013*). However, in our study, we observed a poor relationship between the accumulated rainfall with A(H3N2) and influenza virus B on 1 year scales, which is consistent with the findings in Egypt in which the infection of H5N1 avian influenza was negatively associated with precipitation between 2006 and 2008 (*Monamele et al., 2017*; *Murray & Morse, 2011*). In Jinan, the rainfall season is mainly concentrated in summer with the average temperature >30 °C and dramatically decreases the half-time of influenza virus and transmission capacity. However, we did not observe a relationship between influenza virus and the sunshine duration in our study, which is different from the study of *Yu et al. (2013)*, in which sunshine duration is a key environmental variable in influenza virus transmission.

MWC analysis indicated that the combination of any two of four climate factors influenced the seasonality of different influenza virus types or subtypes. Temperature and atmospheric pressure might be the main influencing factors of A(H3N2) and influenza virus B, whereas temperature and AH might best shape the seasonality of A(H1N1)pdm09. Therefore, we conclude that the influenza seasonality may be influenced by some of the relative climate factors, not by one single climate factor (*Tamerius et al., 2011*). In 2014–2015 influenza epidemic seasons, atmospheric pressure, temperature and RH influenced the seasonality of A(H3N2) and A(H1N1)pdm09; whereas atmospheric pressure, temperature, AH and RH influenced the seasonality of influenza virus B and A(H1N1)pdm09 in the 2015–2016 influenza epidemic seasons.

The study had some interesting findings. First, the dynamic variation of influenza was clear with a peak time in winter season in Jinan, which provided the scientific support for further measure to prevent and control influenza virus, including vaccination time, the time of taking measure by health committee and the high-risk populations.

Second, atmospheric pressure may influence the activity of influenza virus. Previous studies seldom focused on the effect of atmospheric pressure on influenza. In our study, we concluded that atmospheric pressure was positively correlated with three influenza virus types or subtypes, which was used for the influenza virus prediction in the future analysis.

Third, different combination of climate factors may influence different influenza viruses, which further predict the seasonality of different influenza viruses. Spearman's correlation analysis and WTC demonstrated that temperature, AH and atmospheric pressure were main influencing factors and the strength of different combination of climate factors contributed to the seasonality of different influenza virus, which provided a new idea for influenza prevention and control.

## CONCLUSIONS

In all, based on the wavelet analysis, we found the dynamic variations in influenza virus characterized by annual cycle with remarkable winter seasonality and exploited the potential impacts of climate parameters on influenza virus: temperature and atmospheric pressure might be the main influencing factors of A(H3N2) and influenza virus B, whereas temperature and AH might best shape the seasonality of A(H1N1)pdm09. Wavelet analysis indicated that changes in different climate factors lead to different extents of delays in influenza virus occurrence. All of this provides scientific support for influenza virus prevention and control.

## ACKNOWLEDGEMENTS

We are grateful to the staff of the influenza surveillance sentinel hospitals.

### Funding

The study is supported by Health and Family Planning Commission of Shandong Province (2016WS0382 and 2016WS0381). The funders had no role in study design, data collection and analysis, decision to publish, or preparation of the manuscript.

### Grant Disclosures

The following grant information was disclosed by the authors:
Health and Family Planning Commission of Shandong Province: 2016WS0382 and 2016WS0381.

### Competing Interests

The authors declare that they have no competing interests.

### Author Contributions

- Wei Su conceived and designed the experiments, analyzed the data, prepared figures and/or tables, authored or reviewed drafts of the paper, and approved the final draft.
- Ti Liu conceived and designed the experiments, analyzed the data, prepared figures and/or tables, authored or reviewed drafts of the paper, and approved the final draft.
- Xingyi Geng performed the experiments, authored or reviewed drafts of the paper, data collected, and approved the final draft.
- Guoliang Yang performed the experiments, prepared figures and/or tables, and approved the final draft.

**Data Availability**

Raw data is available as a Supplemental File.

**Supplemental Information**

Supplemental information for this article can be found online at http://dx.doi.org/10.7717/peerj.8626#supplemental-information.

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
