# Peer review of "Seasonal pattern of influenza and the association with meteorological factors based on wavelet analysis in Jinan City, Eastern China, 2013–2016"

_PeerJ, doi:10.7717/peerj.8626_

## Round 0.1 · original submission · Major Revisions

You have the opinion of two expert reviewers, each of whom expressed major concerns. You will need to address these if you choose to revise and resubmit.

Both of the reviewers noted the submission requires English-language editing, a conclusion that I explicitly endorse along with their scientific critiques.

Please note that because both reviewers were in agreement on major revisions, I must absolutely stress that acceptance cannot be guaranteed. At the same time, if I did not think there is a path to acceptance then I would recommend rejection to the journal at this time, and I have not done that.

·

Basic reporting

In general, the English language should be improved. The current phrasing makes comprehension very difficult. Some examples where the writing needs to be developed include lines 72, 184, 248, 251, 258, 265, and 299-304.

The article aims to explore the association between several meteorological factors and influenza spread and infection. There are some suggestions spread in the discussion about the mechanisms by which these factors could lead to particular influenza patterns, but since it is a central issue in the manuscript, there should be a separate section in the introduction as a context.

The structure of the article seems to conform to an acceptable format of ‘standard sections.’ The submission appears ‘self-contained.’

Experimental design

The knowledge gap that is intended to fulfill in this analysis is not clearly defined in the manuscript. It is not clear whether the main aim was to describe the synchrony in the timing of influenza epidemics and meteorological factors or to find causal relations between them. Both intentions are stated.

My main concern regarding the analysis strategy is that the study makes explanatory conclusions from bivariate analyses, comparing meteorological variables that are highly correlated, and omitting several pertinent variables. Meteorology is not my specialty, but intuitively and looking at the provided data (Fig. 1), the correlation factors in Table 1, and at some basic literature, meteorological factors as humidity, temperature, atmospheric pressure, wind speed, rainfall, and sunshine duration are highly interdependent. Any separate analysis would mask mediation and confounding mechanisms.
Furthermore, influenza virus subtype (e.g., H1N1, H3N2, B, etc.), that have shown to have considerable influence on the spread and severity of influenza epidemics are omitted from the analysis. This omission is surprising since the analyses are done using laboratory-confirmed influenza information. It is possible that the information at the local level is not available (something that is not mentioned in the paper), in which case other geographical scales could have been used as proxies for virus subtype circulation.
The analyses using the cross wavelet transform and wavelet coherence have the same problem as the bivariate correlation analyses also employed. Since most meteorological phenomena are interdependent, there is no way to identify the real association by restricting the analysis exclusively to bivariate comparisons of seasonal patterns. My thought on this is that these are only measures of synchrony in timing, but they are not able to establish causal relationships.
Although statistical measures and sensitivity analyses can justify such bivariate analyses, there is no justification for the study design employed in this case. As an example, other studies cited by the authors (Yu et al. 2013 and Tamerius et al. 2019) use more robust multivariate analyses, including meteorological variables, as well as influenza virus subtype circulation measures to identify associations between meteorological factors and influenza patterns.

The description in the method section is confusing in several parts. For instance, the description of the construction of the arrows is not well described (lines 139-140). It is not clear about which arrows are being described.

The description, labeled, and aesthetic of the Figures should be substantially improved to conform to an acceptable standard. Some examples where the Figures need to be improved include:
In Figure 1, several kinds of plots are used (bars and lines), and some of the measures are combined without apparent reason (atmospheric pressure and wind speed). I suggest unifying the plot aesthetic when possible and combine measures that are more related, as mean and minimum temperature, for instance. Also, units for the Y-axis should be added.
Captions in Figures 2 and 3 are switched.
X- and Y-axis in wavelet plots (Figures 2 and 3) are not correctly labeled. The x-axis should be something like “time-period (weeks)” and Y-axis something like “frequencies or scale (weeks),” as stated in the captions.
Less imperative but still relevant, I suggest using another palette of colors than rainbow for a continuous scale, as in the case of wavelets. The rainbow colors palette has several drawbacks that have been pointed out elsewhere. For instance, it can induce the perception of ruptures where there is continuity and can make real ruptures appear as a continuous gradient. Some further details: https://www.scientificamerican.com/article/end-of-the-rainbow-new-map-scale-is-more-readable-by-people-who-are-color-blind/?redirect=1
https://www.mathworks.com/tagteam/81137_92238v00_RainbowColorMap_57312.pdf

Validity of the findings

The data on which the conclusions are based is provided.

As mentioned above, the analysis design performed does not allow inferring causality between climatic factors and influenza infection. In consequence, several statements suggesting this kind of relation throughout the text should be avoided and nuanced. Some examples of these expressions are: “We found weekly change in climate variables significantly contributed to dynamic variation of influenza virus” (line 244), “we also found that the influenza seasonality was determined by some of the relative climatic factors” (line 297), “[climatic factors] played important role in shaping influenza seasonality” (line 309-310), “led to” (line 311), and “[we] exploited the impacts of climatic parameters on influenza virus”.
As stated in the guidelines of the journal, “Speculation is welcome, but should be identified as such.”

Additional comments

Besides the points highlighted above, some minor issues:
In the data section, it is stated that influenza-like illness is used in the analysis, but the only influenza information used is laboratory-confirmed influenza.
Maybe I am reading badly the statistical significance in the wavelet plots, but in lines 169 and 175 it is said that the frequencies in the 32-64 week band are not statistically significant, but the black line indicates the opposite.
Line 198, it is written 214 but is 2014.
In line 216 it should be Fig. 3F, not 3G.
In line 216 it should be Fig. 3F, not 3G.

·

Basic reporting

• English language: the paper has a lot of mistakes and definitely need to be reviewed. I tried to help by making notes about things that need to be changed and I will send the scanned version with my notes. The transcription here following the guidelines in impossible, because would require to re-write almost the full paper. I strongly recommend the use of Grammarly or a similar app to help with the grammar and phrasing.
• Literature references: I missed the results from China in the literature review. As the paper is from a specific Chinese city, it would be important to look at findings from other places.
• References: Please check the references. The webpage cited is in incorrect format (page 85). Some other citations are also wrong. For example: please use Tamerius et al (YEAR).
• Figures:
o Figure 1. Please, use graphs of one vertical axis. AH has a different scale than Wind speed, please create 1 graph to each variable. Specify in the figure’s legend the frequency of the data.
o Figure 2: It would be better if we have more dates on the horizontal axis.

Experimental design

It is not an experimental design study.

Validity of the findings

Specific commentaries:
LINES 78-82:
• The outcome needs to be described more clearly. What does it contemplate? The reading of these lines makes me feel that the outcome is ILI + confirmed cases. But in posterior sections, it was not confirmed.
• Why the outcome is measured weekly and annually? If we have weekly data, and it is the most frequent date you have, we have also annual data right?
• It is not clear the frequency of the data you are using
• Do you have missing data? How did you treat it? Imputation? If yes, please refer to that here.
• Please cite reference for the Jinan CDC at the first time you mention it
• You are here presenting your data, so please specify here the type of data you will analyze (positive rate).
• It is important to mention the coverage of the 3 sentinels. Are they all the possible health facility to people looking for care? If it isn’t please specify its coverage.
LINE 87: is the rainfall accumulated? If all meteorological variables are average, please suppress this information, not repeating it each variable is presented. You can say: “we used average measures of…”
LINE 83-89: What was the procedure used to match up the climatic to the influenza data? Both data sets are in different frequencies. Please, let it clear.
Moreover, where is located the meteorological station? Capital? Did you use data from the one closest to the sentinels?
LINES 91-96: Please, be more clear to the chronological analysis: 1) Outcome, 2) Descriptive (include correlation and normality test here), 3) Wavelet analysis.
Important: you didn’t explain to the reader why are you looking to correlation, normality test, and wavelet decomposition. It is important to briefly explain why you are using these methods.
LINES 103-117: I don’t think you have a section here. In general, researched introduce the continuous wavelet in the wavelet analysis. So, I recommend to summarize and join this section with the previous one (Wavelet analysis). Here, the most important information are the “continuous decomposition”, the wavelet mother used and the coherence statistic. In my opinion, all formula could be suppressed, but the concept of the mother wavelet and coherence should be well described.
LINE 113: you have here a citation in another format (no 28).
LINES 118-145: Again, I would suppress the formulas, and explain clearly the coherence and what the reader would expect to find in the power spectrum graphs: the area where the entropy is high, the meaning of the arrows direction and the edge. It is important also to mention the null and alternative hypotheses used to build the confidence intervals: ARIMA? IID? Seasonal and stationary?
LINE 157: Why are you testing for normality? Please explain.
LINE167: “Influenza virus presented an annual epidemic cycle…”
ALL TEXT: please, use: climate factors, not climatic factors
LINES 173-179: I would recommend you to compare the average spectrum power in the periods you are inside the edge. The average power as a whole can be diminished by insignificance from the area outside the edge. Figure 2 seems to corroborate the hypothesis that you have similar annual seasonality in all climate variables and influenza.
LINES 180-218: Please, be specific. More than tell the reader if a climate variable is leading, you should say in how many weeks the peak or minimum values are leading the influenza peak or minimum values. Without this information, you can’t establish a connection between the variables. The fact that (for example) a peak of a climate variable is not leading influenza’s peak does not mean that they are not related. To talk about a lack of relationship we should have no explanation by incubation time, virus spread time, etc.

---

## Round 0.2 · Minor Revisions

Your revised manuscript is much improved in the eyes of both reviewers. Thank you for your diligence on the first round of revisions. The second round of revisions, which I propose now, should be more straightforward.

·

Basic reporting

The text is clear and unambiguous.

There are some small typos, for instance, in lines 294, 319, 321, 328.

There are problems with the captions of Figures 8 and 9. The virus subtype mentioned in the description is not the correct one.

The link provided for the Mathlab package in line 142 does not work. I also think that besides the link, the authorship of the package (Grinsted) should be better cited in the manuscript as a reference

The color scale in the wavelet plots is not pink to white. It seems more like dark-brown to white.

Change "atmosphere pressure" for "atmospheric pressure" in lines 352-353.

Experimental design

It would be very informative to include analyses of all-subtype influenza and compare them with the analyses by subtype. Since subtypes are competing with each other, this could affect the seasonal pattern of each subtype.

Validity of the findings

no comment

Additional comments

The quality of the paper improved considerably since the first version. Most of the issues signaled in the previous review were adjusted.

I highlight two aspects that I consider need to be addressed:

The differences across virus subtypes in the associations between climate factors and seasonality are an interesting finding. In addition to the suggestion made earlier about including analyses of all-subtype influenza, I find that it is imperative to mention if there is a possible interpretation of these differences across subtypes. Its is not clear for me whether these differences were already noticed in previous work, or this is the first study to report them.

My main concern with the previous version of the manuscript was the fact that statistically significant associations and co-movements were interpreted as causal relationships. I find that this issue is very well addressed in the discussion of the new version of the paper, by clarifying that these are potential relationships suggested by the theory and results. However, at the end of the discussion (lines 339-342) and in the conclusions, there are still strong sentences indicating causality, by using verbs such as “influenced” and “shaped.” I suggest employing more cautious statements for interpreting the findings, as it was done in most of the discussion section.

·

Basic reporting

The manuscript has improved a lot since the last version.

Basic Reporting
· English language & formating: the language improved a lot, but we still can find some grammar mistakes. I am sending my notes scanned, in case it helps.
· Figures: all problems were solved, and the pictures have improved a lot.
Specific commentaries:
· Methods – Influenza surveillance data: How much the three sentinels represent in terms of influenza offer of care? Please specify its coverage.
· LINES 262-263: please make it clear what combination are you referring to.
· LINES 340-343: all the linear correlations obtained are very small. I wouldn’t attribute to influenza B a relationship with temperature if the wavelets didn’t point out to that.
· LINES 364-365: same commentary above.

Experimental design

The research question is original and challenging, but it could have its epidemiological implications better explored in the discussions.
The methods description improved a lot!
Now, it is clear what data they have used, the frequency and statistical procedures throughout the analysis.

Validity of the findings

The authors relied in a lot of methods, and it makes it difficult to have a clear and objective conclusion.
The data and all graphs have been provided, and they are in line with the results and conclusion sessions.

Additional comments

It lacks and explanation to the different sets os variables found as associated with the influenza subtypes. I would invest a little more on that to move the quality of your work one step ahead.

---

## Round 0.3 · Minor Revisions

One of the referees still has some minor critiques if you could address/fix these please

I don't like to go through many rounds of revisions, but I must say that the referee's comments are well-founded, especially when he notes that several assertions "are not supported by the results".

·

Basic reporting

In general, the quality of the text is unambiguous and professional.
However, there are still several aspects that should be adjusted to reach publishable quality. Some examples (not the only ones):
- Through the text, it is used both “climate factors” and “climatic factors” (lines 212, 353).
- Lines 219-220: “In terms of temperature, in the 26-week band 220 (2013–2014), influenza virus A (H1N1) pdm09 was negatively correlated with the left arrows.” This could lead to misinterpretation because there is no correlation “with the left arrows”, but the arrows indicate the negative correlation.
- The word “virus” is overused in some sections. e.g., it is repeated 13 times in lines 269-275.
- Line 164: "Figure 2D", but it actually refers to Figure 2E.
- Lines 215-235: The results described correspond to Figures 5 and 6, but there is no explicit reference to these plots in the text.
- Lines 266 and 341: "in which" instead of "that"
- Line 294: "obtained" is not the appropriate term here.
- Line 297: "lags" instead of "lag"
- Lines 331, 339, 345, should be "a" instead of "the"
- Line 361: not clear what the expression "and people" means.
- Lines 325 and 368: "illuminated" might not be the appropriate term.
- Lines 373: "elaborated" might not be the appropriate term.
- There are several typos in the main text, e.g., spaces lacking in lines 154, 160, 161, 164, 173; other typos in lines 362, 366...

Experimental design

Most of the findings were obtained using wavelet plots, but the components of these plots are not described in the method section. There is no reference to the meaning of the arrows, their direction, the colors, the thick black curves, or the cone of influence in this section.
Part of this information is written in the results section, but it is spread here and there, and part of it seems incorrect, as will be detailed below. The caption of the figures is the only place where this description is done. this description should be better organized and has some substantial contradictions with the information consigned in the results section.
More precisely, in the results section (lines 193-194) is written: “The cone of influence separates the reliable (white colors) and less reliable (pink colors) regions.” In contrast, in the caption of Figure 1 is written "The cone of influence separating the regions with reliable and less reliable estimates is represented by the lighter pale colors. […] The region outside the black-curved cone indicates the presence of edge effects.” Both descriptions are contradictory, and it seems that the latter is the correct one.
As stated in the caption, the scale white-pink is not related to the cone of influence (edge effects) but to the “power” of the correlation (“The color code for power ranges from pink (low power) to white (high power)”).

Validity of the findings

Some expressions are not supported by the results.
- Line 294: "atmospheric pressure influenced the variations in different influenza virus types".
- Line 297: "Furthermore, during some time periods, the prevalence of influenza virus 'lag' the changes in atmospheric pressure, which can predict the occurrence of influenza virus A(H3N2)epidemics."
The study does not provide support to affirm that atmospheric pressure "influence" or "can predict" variations in virus subtype.

---

## Round 0.4 · accepted · Accept

Thank you for your diligence with the revisions and for your patience with the editorial process.